REGISTERED REPORT

# Registered report: Oncometabolite 2-hydroxyglutarate is a competitive inhibitor of α-ketoglutarate-dependent dioxygenases

Brad Evans[1], Erin Griner[2], Reproducibility Project: Cancer Biology*

[1]Proteomics and Mass Spectrometry Facility, Donald Danforth Plant Science Center, St. Louis, Missouri, United States; [2]University of Virginia, Charlottesville, Virginia, United States

**Abstract** The Reproducibility Project: Cancer Biology seeks to address growing concerns about reproducibility in scientific research by conducting replications of selected experiments from a number of high-profile papers in the field of cancer biology. The papers, which were published between 2010 and 2012, were selected on the basis of citations and Altmetric scores (*Errington et al., 2014*). This Registered report describes the proposed replication plan of key experiments from 'Oncometabolite 2-hydroxyglutarate is a competitive inhibitor of α-ketoglutarate-dependent dioxygenases' by Xu and colleagues, published in *Cancer Cell* in 2011 (*Xu et al., 2011*). The key experiments being replicated include Supplemental Figure 3I, which demonstrates that transfection with mutant forms of IDH1 increases levels of 2-hydroxyglutarate (2-HG), Figures 3A and 8A, which demonstrate changes in histone methylation after treatment with 2-HG, and Figures 3D and 7B, which show that mutant IDH1 can effect the same changes as treatment with excess 2-HG. The Reproducibility Project: Cancer Biology is a collaboration between the Center for Open Science and Science Exchange, and the results of the replications will be published by *eLife*.

*For correspondence: fraser@ scienceexchange.com

Group author details
Reproducibility Project: Cancer Biology
See page 26

## Introduction

Mutations in IDH1 and IDH2 are found in gliomas and in acute myeloid leukemia. All mutations are heterozygous and result in changes to one of two amino acids: arginine 132 in IDH1, or either arginine 172 or arginine 140 in IHD2. Wild-type IDH1 catalyzes the conversion of isocitrate to α-ketoglutarate (α-KG). The arginine mutations abolish its normal activity and instead mutant IDH1 and IDH2 reduce α-KG to generate the oncometabolite 2-hydroxyglutarate (2-HG) (*Ward et al., 2010*), which in turn affects the function of multiple α-KG dependent dioxygenases, including the TET family of 5-methylcytosine (5 mC) hydroxylases (*Kinney and Pradhan, 2012*; *McKenney and Levine, 2013*). In their *Cancer Cell* 2011 paper, Xu and colleagues examined the effects of excess production of 2-HG on downstream processes that could affect cancer progression. They showed that 2-HG could act as a competitive inhibitor for α-KG-dependent DNA demethylases, specifically Tet2. Ectopic expression of the mutant forms of IDH1 and IDH2 inhibited histone demethylation and 5mC hydroxylation. Examination of glioma samples from patients also showed that mutations in IDH1 were associated with increased histone methylation and decreased 5-hydroxymethylcytosine (5hmC) levels (*Xu et al., 2011*).

In Supplemental Figure 3I, Xu and colleagues demonstrated that transfection of U-87 MG cells with the mutant IDH1[R132H] increased the amount of 2-HG in the cells, as compared to transfection with wild-type IDH1 (*Xu et al., 2011*). This is evidence that mutant IDH1 changes the physiological levels of 2-HG, and is replicated in Protocol 1.

Xu and colleagues first showed that 2-HG can occupy the same binding pocket as α-KG in *Caenorhabditis elegans* KDM7A, indicating it acts as a competitive inhibitor of α-KG. Importantly, they also presented evidence that 2-HG may outcompete α-KG, since 2-HG levels affected many enzymatic functions normally dependent on α-KG. In Figure 3A, they treated U-87 MG cells with cell permeable versions of α-KG and 2-HG, and examined levels of histone methylation by Western Blot. Treatment with increasing amounts of 2-HG led to increases in H3K9me2 and H3K79me2, consistent with the idea that 2-HG inhibited histone demethylases. This effect was abolished by co-treatment with α-KG, confirming a competitive relationship between the two metabolites (*Xu et al., 2011*). This experiment is replicated in Protocol 2. Xu and colleagues also examined the effect of 2-HG on the TET family of 5 mC hydroxylases using an in vitro system of purified TET2 and double-stranded oligos containing a 5mC restriction digestion site in Figure 8A. Adding increasing concentrations of 2-HG abolished the ability of TET2 to convert 5 mC to 5hmC (*Xu et al., 2011*). This experiment will be replicated in Protocol 5.

In addition to demonstrating that the metabolite 2-HG can affect the activity of α-KG-dependent enzymes, Xu and colleagues showed that treatment with mutant forms of IDH1 and IDH2 resulted in similar outcomes. In Figure 3D, they transfected U-87 MG cells with IDH1$^{R132H}$ and assessed levels of histone methylation by Western blot. Transfection with IDH1$^{R132H}$ increased histone methylation, and treatment with α-KG abolished this increase in histone methylation, consistent with the idea that α-KG and 2-HG are competitive metabolites (*Xu et al., 2011*). This experiment will be replicated in Protocol 3. In Figure 7B, they also examined TET activity in the presence of mutant IDH1. While 5hmC levels are normally undetectable in HEK293 cells, transfection with TET catalytic domain (CD)-expressing plasmids increased 5hmC levels to detectable amounts. Co-transfection of TET-CD and wild-type IDH1 or IDH2 increased levels of 5hmC, as expected, while co-transfection of TET-CD with mutant forms of IDH1 and IDH2 decreased 5hmC levels (*Xu et al., 2011*). This experiment is replicated in Protocol 4.

The work of Xu and colleagues (*Xu et al., 2011*), along with work from Figueroa and colleagues (*Figueroa et al., 2010*) and Lu and colleagues (*Lu et al., 2012*), has generated much interest in the role of altered metabolites in the changing methylation patterns seen in various types of cancer. Using a different cell line than Xu and colleagues, Lu and colleagues demonstrated that mutations in IDH2, similar to mutations in IDH1, also generated abnormal levels of 2-HG which correlated with increased global methylation levels (*Lu et al., 2012*). Kernystsky and colleagues, Duncan and colleagues and Turcan and colleague have also shown that expression of exogenous mutated IDH genes in immortalized human cancer cell lines or in erythroid progenitor cells caused increased production of 2HG and increased levels of methylation (*Duncan et al., 2012*; *Turcan et al., 2012*; *Kernytsky et al., 2015*). Sasaki and colleagues extended these inquiries by generating conditional knock-in IDH1 mutant mice. These mice displayed elevated serum levels of 2HG and similar patterns of hypermethylation as observed in AML patients (*Sasaki et al., 2012*). Akbay and colleagues generated IDH2 mutant mice and also observed an increase in global methylation in heart tissue. They also demonstrated that mice carrying IDH mutant xenograft tumors displayed higher serum levels of 2HG (*Akbay et al., 2014*). Recently, 2-HG production has also been associated with MYC activation in some breast cancers, which also displayed increased levels of methylation as compared to tumors with lower levels of 2-HG (*Terunuma et al., 2013*).

## Materials and methods

Unless otherwise noted, all protocol information was derived from the original paper, references from the original paper, or information obtained directly from the authors. An asterisk (*) indicates data or information provided by the Reproducibility Project: Cancer Biology core team. A hashtag (#) indicates information provided by the replicating lab.

### Protocol 1: Gas chromatography-mass spectrometry measurement of cellular α-KG and 2-HG concentrations in U87MG cells ectopically expressing mutant IDH1

This protocol describes how to transfect cells with exogenous wild-type IDH1 or mutant IDH1$^{R132H}$ and assess levels of α-KH and 2-HG by gas chromatography-mass spectrometry (GC-MS), as seen in Supplemental Figure 3I.

## Sampling

- This experiment will be repeated independently 5 times for a final power of at least 92%.
  ○ See Power calculations for details.
- Each experiment consists of three cohorts:
  ○ Cohort 1: U-87 MG cells transfected with vector alone.
  ○ Cohort 1: U-87 MG cells transfected with wild-type IDH1.
  ○ Cohort 1: U-87 MG cells transfected with mutant IDH1.
  Each cohort will be assessed via GC–MS for:
  - $\alpha$-KG levels.
  - 2-HG levels.

## Materials and reagents

| Reagent | Type | Manufacturer | Catalog # | Comments |
|---|---|---|---|---|
| U-87 MG cells | Cells | ATCC | HTB-14 | |
| N-methyl-N-[tert-butyldimethylsilyl] trifluoroacetamide | Chemical | Sigma–Aldrich | 375934-10× 1 M | |
| Agilent 6890-5973 gas chromatograph-mass spectrometer | Instrument | Agilent | 6890-5973 | |
| HP-5MS column | Material | Agilent | 19091S-433I | |
| 60 mm tissue culture dishes | Material | Corning | 430166 | |
| DMEM; high glucose | Medium | Sigma–Aldrich | D5671 | Original unspecified |
| Opti-MEM Reduced Serum Medium | Medium | Life Technologies | 31985-062 | Original unspecified |
| Vector only plasmid (GFP) | Plasmid | Provided by the original authors | | |
| IDH1-IRES-GFP vector | Plasmid | Provided by the original authors | | |
| IDH1$^{R132H}$-IRES-GFP vector | Plasmid | Provided by the original authors | | |
| FBS | Reagent | Sigma–Aldrich | F2442 | Original unspecified |
| Trypsin-EDTA solution, 1× | Reagent | ATCC | ATCC-30-2101 | |
| Penicillin-streptomycin solution | Reagent | ATCC | ATCC-30-2300 | |
| *TransIT*®-LT1 transfection reagent | Reagent | Mirus Bio | MIR 2300 | Replaces SunBio-EZ (SunBio) |
| Methoxyamine hydrochloride | Reagent | Sigma–Aldrich | 226904 | |
| Pyridine | Reagent | Sigma–Aldrich | 33,553 | |
| GenElute Endotoxin-free Plasmid Maxiprep Kit | Kit | Sigma–Aldrich | PLEX15-1KT | |
| $\alpha$-KG | Chemical | Sigma–Aldrich | 75892 | |
| L-2-HG | Chemical | Sigma–Aldrich | 90790 | |
| 0.2 μm filter vials | Material | Restek | 25893 | |
| Centrivap | Equipment | Labonco | | |
| Anti-GAPDH-HRP | Antibody | Abcam | ab9385 | |
| Mouse monoclonal IgG$_1$ $\alpha$ IDH1 | Antibody | Abcam | ab117976 | The original catalog number was not specified |

## Procedure

### Notes

- U-87 MG cells are maintained in DMEM supplemented with 10% FBS at 37°C/5% $CO_2$.
- All cells will be sent for mycoplasma testing and STR profiling.

1. Transform, grow up and maxiprep vector only (GFP), IDH1-IRES-GFP, and IDH1$^{R132H}$-IRES-GFP plasmids using a Endo-free Maxiprep kit following manufacturer's instructions.
   a. Confirm plasmid identity by sequencing.

2. Plate U-87 MG cells in 60 mm dishes.
   a. First, run an optimization to determine the growth rate of the cells and optimal number of cells per plate for transfection.
3. 24 hr after plating, transfect cells with plasmids using *Trans*IT-LT1 transfection reagent and Opti-MEM medium according to the manufacturer's protocol.
   a. Transfect 8 µg of DNA per construct using appropriate amount of transfection reagent.
      i. Vector only (GFP).
      ii. Wild-type IDH1 (IDH1-IRES-GFP).
      iii. Mutant IDH1 (IDH1$^{R132H}$-IRES-GFP).
   b. Prepare two plates per cohort; one will be harvested for Western blot confirmation of protein expression (Step 4), the other will be used for metabolite analyses (Step 5).
4. For Western blot: 48 hr after transfection, confirm protein expression by Western blot. Note: perform each time cells are transfected.
   a. Run Western blot as outlined in Protocol 2 Steps 3 through 17 with the following modifications:
      i. Blots do not need to be stripped and re-probed.
      ii. Blots will be probed with:
         1. Anti-IDH1; diluted according to the manufacturer's recommendation.
         2. Anti-GAPDH-HRP; 1:5000.
            a. Loading control.
5. For metabolite analysis: 24 hr after transfection, remove culture medium, wash cells with cold PBS and immediately add 10 ml of pre-chilled (−80°C) 80% (vol/vol) methanol. Harvest cells by scraping and lyophilize following the manufacturer's instructions.
   a. Samples will be lyophilized in a speedvac with no heating to keep samples frozen throughout. Immediately after drying remove samples from the speedvac for derivitization.
6. Oximate lyophilized samples with 20 µl 20 mg/ml methoxyamine hydrochloride in pyridine at 30°C for 60 min.
7. Derivatize samples for 30 min at 70°C in 80 µl pyridine and 20 µl N-methyl-N-[tert-butyldimethylsilyl] trifluoroacetamide.
8. Filter samples using 0.2 µm filter vials (PTFE).
9. Inject 3 µl of samples for gas chromatography-mass spectrometry analysis (GC–MS) into Agilent 6890-5973 GC–MS. Use a HP-5MS column (30 m 0.25 mm 0.25 µm) for analysis. Program GC oven temperature from 60°C to 180°C at 5°C/min and from 180°C to 260°C at 10°C/min. Set the flow rate of carrier gas at 1 ml/min. Operate the mass spectrometer in the electron impact (EI) mode at 70 eV.
10. Calculate relative α-KG and 2-HG concentrations by normalizing α-KG (29.86 min) and 2-HG (30.10 min) peak areas to the average of L-threonine (29.58 min), L-serine (29.96 min) and L-phenylalanine (30.74 min) peak areas.
11. Repeat independently four additional times.

## Deliverables

- Data to be collected:
  ○ Chromatograms and sequence files confirming plasmid identity.
  ○ Data generated determining growth optimization.
  ○ Full image of Western blot showing protein expression and loading controls.
  ○ Mass spectra readouts of all samples.
  ○ Raw values of peak areas for α-KG (29.86 min), 2-HG (30.10 min), L-threonine (29.58 min), L-serine (29.96 min) and L-phenylalanine (30.74 min).
  ○ Quantification of average of peak areas for L-threonine (29.58 min), L-serine (29.96 min) and L-phenylalanine (30.74 min).
  ○ Quantification of relative α-KG and 2-HG concentrations by normalization to average peak areas for L-threonine (29.58 min), L-serine (29.96 min) and L-phenylalanine (30.74 min).
  ○ Bar graphs of relative α-KG or 2-HG concentrations (in percent) for each cell line (as in Supplemental Figure 3I).

## Confirmatory analysis plan

- Statistical analysis of the replication data:

○ Note: At the time of analysis we will perform the Shapiro–Wilk test and generate a quantile–quantile plot to assess the normality of the data. We will also perform Levene's test to assess homoscedasticity. If the data appears skewed we will perform the appropriate transformation in order to proceed with the proposed statistical analysis. If this is not possible we will perform the equivalent non-parametric test.
○ One-way MANOVA of α-KG and 2-HG levels in vector-transfected, IDH1-wildtype transfected, and IDH1$^{R132H}$-transfected cells with the following Bonferroni corrected comparisons:
  ▪ α-KG levels planned comparisons:
    • vector vs IDH1$^{WT}$.
    • vector vs IDH$^{R132H}$.
    • IDH1$^{WT}$ vs IDH$^{R132H}$.
  ▪ 2-HG levels planned comparisons:
    • vector vs IDH1$^{WT}$.
    • vector vs IDH$^{R132H}$.
    • IDH1$^{WT}$ vs IDH$^{R132H}$.
- Meta-analysis of original and replication attempt effect sizes:
  ○ Compute the effect sizes of each comparison, compare them against the reported effect size in the original paper and use a meta-analytic approach to combine the original and replication effects, which will be presented as a forest plot.

## Known differences from the original study

- Aspects of the Western blot protocol are provided by the replicating lab; complete details of the original protocol were unavailable.
- Since the cell density during transfection is unknown in the original paper, the replicating lab will optimize growth conditions and cell density for transfection.

## Provisions for quality control
All data obtained from the experiment—raw data, data analysis, control data and quality control data—will be made publicly available, either in the published manuscript or as an open access dataset available on the Open Science Framework (https://osf.io/kvshc/).

- Sequence data confirming plasmid identity.
- Western blots confirming exogenous protein expression.
- STR profiling confirming cell line authenticity.
- Mycoplasma testing confirming lack of contamination.
- Growth characteristics of the cells will be optimized.

## Protocol 2: Western blot to assess histone methylation in U-87 MG cells following treatment with oct-2-HG and/or oct-α-KG

This protocol describes how to treat U-87 MG cells with cell permeable versions of 2-HG and α-KG and assess histone methylation via Western blot, as seen in Figure 3A and Supplemental Figure 3F.

### Sampling

- The experiment will be repeated independently 3 times for a final power of 84%.
  ○ See Power calculations for details.
- Each experiment consists of four cohorts:
  ○ Cohort 1: untreated U-87 MG cells.
  ○ Cohort 2: U-87 MG cells treated with 10 mM racemic Oct-2-HG.
  ○ Cohort 3: U-87 MG cells treated with 20 mM racemic Oct-2-HG.
  ○ Cohort 4: U-87 MG cells treated with 20 mM racemic Oct-2-HG and 5 mM oct-α-KG.
  Each sample will be blotted for:
    ▪ H3K9me2.
    ▪ H3K79me2.
    ▪ H3.

## Materials and reagents

| Reagent | Type | Manufacturer | Catalog # | Comments |
|---|---|---|---|---|
| Mouse monoclonal anti-H3K9me2 | Antibody | Abcam | Ab1220 | The original catalog number was not specified |
| Mouse monoclonal anti -H3K79me2 | Antibody | Abcam | Ab3594 | The original catalog number was not specified |
| Mouse monoclonal anti -H3 | Antibody | Abcam | ab10799 | The original catalog number was not specified |
| Goat Anti-Mouse IgG H&L (HRP) | Antibody | Abcam | ab97023 | We will use this for all mouse primaries |
| U-87 MG cells | Cells | ATCC | HTB-14 | |
| 60 mm tissue culture dishes | Material | Corning | 430166 | |
| DMEM; high glucose | Medium | Sigma–Aldrich | D5671 | Original unspecified |
| FBS | Reagent | Sigma–Aldrich | F2442 | Original unspecified |
| Oct-$\alpha$-KG | Reagent | Cayman Chemical | 11970 | |
| 2S(L)-Oct-2-HG | Reagent | TRC | H942596 | Original synthesized in house |
| 2R(L)-Oct-2-HG | Reagent | TRC | H942595 | |
| Protease inhibitor cocktail (mammalian) | Reagent | Sigma–Aldrich | P8340-1ML | Original not specified |
| TruPAGE TEA-tricine SDS running buffer (20×) | Reagent | Sigma–Aldrich | PCG3001-500 ML | Original not specified |
| TruPAGE LDS sample buffer (4×) | Reagent | Sigma–Aldrich | PCG3009-10 ML | Original not specified |
| TruPAGE DTT sample reducer (10×) | Reagent | Sigma–Aldrich | PCG3005-1ML | Original not specified |
| TruPAGE transfer buffer (20×) | Reagent | Sigma–Aldrich | PCG3011-500 ML | Original not specified |
| PBS, without $MgCl_2$ and $CaCl_2$ | Reagent | Sigma–Aldrich | D8537 | Original not specified |
| Hybond ECL nitrocellulose membranes; 20 cm × 20 cm | Reagent | GE Healthcare (Sigma–Aldrich) | GERPN2020D | Original not specified |
| Ponceau S solution; 0.1% (wt/vol) in 5% acetic acid | Reagent | Sigma–Aldrich | P7170 | Original not specified |
| Tris Buffered Saline (TBS); 10× solution | Reagent | Sigma–Aldrich | T5912 | Original not specified |
| Bradford reagent | Reagent | Sigma–Aldrich | B6916 | Original not specified |
| ECL DualVue Western Blotting Markers | Reagent | GE Healthcare (Sigma–Aldrich) | GERPN810 | Original not specified |
| ECL Prime Western blotting system | Reagent | GE Healthcare (Sigma–Aldrich) | GERPN2232 | Original not specified |
| ImageQuant | Software | Molecular Dynamics | Version 5.2 | |
| Typhoon scanner | Equipment | GE Healthcare | | |

## Procedure

### Notes

- U-87 MG cells are maintained in DMEM supplemented with 10% FBS at 37°C/5% $CO_2$.
- All cells will be sent for mycoplasma testing and STR profiling.

1. Plate U-87 MG cells in 60 mm dishes.
2. 24 hr after plating, treat cells with 10 or 20 mM racemic Oct-2-HG or 5 mM Oct-$\alpha$-KG or vehicle (DMSO) for 4–6 hr.
   a. To form racemic mixtures of Oct-2-HG, mix equal amounts of the L and R enantiomers.
3. Wash cells once with cold PBS, then lyse cells in 0.5 mL of SDS loading buffer.
   a. 4× SDS-PAGE loading buffer: 50 mM Tris pH 6.8, 2% SDS, 10% glycerol, 1% B-ME, 12.5 mM EDTA, 0.02% bromophenol blue.
   b. #Measure protein concentration using a CBX assay.
4. Heat lysates at 99°C for 10 min.
5. Run equal amounts of protein per well on a 4–20% SDS-PAGE gel at 220V until ladder marker reaches the bottom of the gel.
6. #Equilibrate gel in transfer buffer for 15 min.
7. #Meanwhile, cut membrane and 4 pieces 3 MM filter paper to size of gel.

a. Soak membrane in MeOH for a few seconds, then wash with $H_2O$.
b. Soak membrane, 3 MM filter paper and pads in transfer buffer.
    i. Transfer buffer: 38 mM glycine, 47 mM Tris, 11 mM SDS, 20% MeOH.
8. #Assemble transfer cassette:
a. red pole (+) < clear plate < pad < 2 × 3 MM filter paper < membrane < gel < 2 × 3 MM filter paper < pad < black pole (−).
9. #Add stirring bar and ice box to transfer box and fill box with transfer buffer until cassette is submerged.
a. Run at 100 V for 1 hr.
10. #Wash membrane in wash buffer for 2 × 5 min.
a. Wash buffer: 1× PBS with 0.05% Tween-20 and 0.1% sodium azide.
11. #Incubate membrane in blocking buffer for 30 min.
a. Blocking buffer: 3% non-fat milk in PBS.
12. #Incubate membrane with one of the following primary antibody in blocking buffer for 2 hr at RT or O/N at 4°C (use manufacturer's suggested dilution in blocking buffer).
a. H3K9me2.
b. H3K79me2.
c. H3.
    i. See Step 17 to strip and re-probe the blot with subsequent antibodies.
13. #Wash 5 min 2× with wash buffer.
14. #Incubate membrane with secondary antibody for 90 min at RT (use manufacturer's suggested dilution in blocking buffer).
a. HRP-conjugated Goat Anti-Mouse IgG H&L: 1: 2000.
15. #Wash 3 × 5 min in wash buffer.
16. #Detect HRP-conjugated secondary antibodies with chemiluminescent detection according to the manufacturer's protocol and image on the Typhoon scanner.
17. Strip the blot in between probes:
a. Wash the membrane with 100 ml stripping buffer (100 mM beta-mercaptoethanol, 1% SDS 25 mM glycine pH 2.0) for 30 min with agitation.
b. Wash the stripped membrane twice with Western blotting wash buffer, 600 ml each wash, for 10 min with agitation.
c. Go to the blocking step of the western blot protocol.
d. Check that stripping was successful by repeating the detection step (without re-probing). Record image of the stripped gel. This will confirm the first antibody-HRP conjugate is removed and/or inactivated. If the stripping procedure is successful, wash the membrane with washing buffer and repeat the blocking-probing and detection steps for the second antibody.
    i. Note: if stripping is unsuccessful, individual blots will be performed.
18. Quantify intensity of bands on western blots using ImageQuant 5.2. Normalize H3k9me2 and H3K79me2 values to total H3 protein level.
19. Repeat independently 2 additional times.

## Deliverables

- Data to be collected:
  ○ Full scans of western blots for H3K9me2, H3K79me2 and H3 including ladder.
  ○ Raw values of intensity of western blot bands.
  ○ Quantification of H3K9me2 or H3K79me2 values normalized to total protein level. Levels of H3K9me2 and H3K79me2 in vehicle treated cells are set to relative intensity = 1 and all other conditions are expressed as fold change relative to the values for vehicle treated cells.
  ○ Quantification of average values and standard deviations for each condition for triplicate experiments.
  ○ Bar graph of average ± standard deviation of H3K9me2 and H3K79me2 levels normalized to H3 for each condition. Fold change in intensity relative to vehicle treated cells is plotted on the y axis (as seen in Supp. Figure 3F).

## Confirmatory analysis plan

- Statistical analysis of the replication data:
  ○ Note: At the time of analysis we will perform the Shapiro–Wilk test and generate a quantile–quantile plot to assess the normality of the data. We will also perform Levene's test to assess homoscedasticity. If the data appears skewed we will perform the appropriate transformation in

order to proceed with the proposed statistical analysis. If this is not possible we will perform the equivalent non-parametric test.
  ○ One-way MANOVA of normalized H3K9me2 and H3K79me3 levels in U-87 MG cells untreated or treated with 10 mM Oct-2-HG, 20 mM Oct-2-HG, or 20 mM Oct-2-HG and 5 mM alpha-KG with the following Bonferroni corrected comparisons:
    ■ H3K9me2 planned comparisons:
      • 0 mM 2-HG vs 10 mM 2-HG.
      • 0 mM 2-HG vs 20 mM 2-HG.
      • 20 mM 2-HG vs 5 mM α-KG + 20 mM 2-HG.
    ■ H3K79me3 planned comparisons:
      • 0 mM 2-HG vs 10 mM 2-HG.
      • 0 mM 2-HG vs 20 mM 2-HG.
      • 20 mM 2-HG vs 5 mM α-KG + 20 mM 2-HG.
- Additional statistical analysis for comparison to the original reported data:
  ○ Bonferroni corrected one-sample *t*-tests of normalized H3K9me2 levels of the following conditions compared to 1 (0 mM 2-HG):
    ■ 10 mM 2-HG.
    ■ 20 mM 2-HG.
  ○ Bonferroni corrected one-sample *t*-tests of normalized H3K79me3 levels of the following conditions compared to constant (0 mM 2-HG set to 1):
    ■ 10 mM 2-HG.
    ■ 20 mM 2-HG.
- Meta-analysis of original and replication attempt effect sizes:
  ○ Compute the effect sizes of each comparison, compare them against the reported effect size in the original paper and use a meta-analytic approach to combine the original and replication effects, which will be presented as a forest plot.

## Known differences from the original study

- The original racemic mixture of Oct-2-HG was synthesized in house by the original lab. The replicating lab is purchasing both L and R enantiomers and mixing them in equal amounts to form a racemic mixture.
- Aspects of the Western blot protocol are provided by the replicating lab; complete details of the original protocol were unavailable.

## Provisions for quality control

All data obtained from the experiment—raw data, data analysis, control data and quality control data—will be made publicly available, either in the published manuscript or as an open access dataset available on the Open Science Framework (https://osf.io/kvshc/).

- STR profiling confirming cell line authenticity.
- Mycoplasma testing confirming lack of contamination.
- Images of stripped gel membranes confirming stripping was successful.

## Protocol 3: Transfection of U-87 MG cells and determination of histone methylation by western blot

This protocol describes the transfection of U-87 MG cells with the mutant form of IDH1 and assessing methylation by Western blot, as seen in Figure 3D and Supplemental Figure 3J.

### Sampling

- This experiment will be repeated independently 6 times for a final power of 94%.
  ○ See Power calculations for details.
- Each experiment consists of 5 cohorts:
  ○ Cohort 1: untransfected cells [additional control].
  ○ Cohort 2: Vector transfected cells [additional control].
  ○ Cohort 3: Vector transfected cells + vehicle.
  ○ Cohort 4: IDH1$^{R132H}$ transfected cells + vehicle.
  ○ Cohort 5: IDH1$^{R132H}$ transfected cells + 5 mM oct-α-KG.
  ○ Each cohort is probed with antibodies against:

- H3.
- IDH1.
- H3K4me1.
- H3K4me3.
- H3K9me2.
- H3K27me2.

## Materials and reagents

| Reagent | Type | Manufacturer | Catalog # | Comments |
|---|---|---|---|---|
| Mouse monoclonal IgG$_3$ α H3 | Antibody | Abcam | ab10799 | The original catalog number was not specified |
| Mouse monoclonal IgG$_1$ α IDH1 | Antibody | Abcam | ab117976 | The original catalog number was not specified |
| Rabbit α H3K4me1 | Antibody | Abcam | ab8895 | The original catalog number was not specified |
| Mouse monoclonal IgG$_{2b}$ α H3K4me3 | Antibody | Abcam | ab6000 | The original catalog number was not specified |
| Mouse monoclonal IgG$_{2a}$ α H3K9me2 | Antibody | Abcam | ab1220 | The original catalog number was not specified |
| Rabbit α H3K27me2 | Antibody | Abcam | ab24684 | The original catalog number was not specified |
| Rabbit α H3K79me2 | Antibody | Abcam | ab3594 | The original catalog number was not specified |
| U-87 MG cells | Cells | ATCC | HTB-14 | |
| 60 mm tissue culture dishes | Material | Corning | 430166 | Or equivalent |
| DMEM; high glucose | Medium | Sigma–Aldrich | D5671 | Original unspecified |
| FBS | Reagent | Sigma–Aldrich | F2442 | Original unspecified |
| Empty vector plasmid | Plasmid | Provided by original authors | | |
| IDH1$^{R132H}$ expression vector | Plasmid | Provided by original authors | | |
| TransIT®-LT1 transfection reagent | Reagent | Mirus Bio | MIR 2300 | Replaces SunBio-EZ (SunBio) |
| Oct-α-KG | Reagent | Cayman Chemical | 11970 | |
| Typhoon scanner | Equipment | GE Healthcare | | |
| ImageQuant | Software | Molecular Dynamics | Version 5.2 | |
| Protease inhibitor cocktail (mammalian) | Reagent | Sigma–Aldrich | P8340-1ML | Original not specified |
| TruPAGE TEA-tricine SDS running buffer (20×) | Reagent | Sigma–Aldrich | PCG3001-500 ML | Original not specified |
| TruPAGE LDS sample buffer (4×) | Reagent | Sigma–Aldrich | PCG3009-10 ML | Original not specified |
| TruPAGE DTT sample reducer (10×) | Reagent | Sigma–Aldrich | PCG3005-1ML | Original not specified |
| TruPAGE transfer buffer (20×) | Reagent | Sigma–Aldrich | PCG3011-500 ML | Original not specified |
| PBS, without MgCl$_2$ and CaCl$_2$ | Reagent | Sigma–Aldrich | D8537 | Original not specified |
| Hybond ECL nitrocellulose membranes; 20 cm × 20 cm | Reagent | GE Healthcare (Sigma–Aldrich) | GERPN2020D | Original not specified |
| Ponceau S solution; 0.1% (wt/vol) in 5% acetic acid | Reagent | Sigma–Aldrich | P7170 | Original not specified |
| Tris buffered saline (TBS); 10× solution | Reagent | Sigma–Aldrich | T5912 | Original not specified |
| Bradford reagent | Reagent | Sigma–Aldrich | B6916 | Original not specified |
| ECL DualVue Western blotting markers | Reagent | GE Healthcare (Sigma–Aldrich) | GERPN810 | Original not specified |
| ECL prime Western blotting system | Reagent | GE Healthcare (Sigma–Aldrich) | GERPN2232 | Original not specified |
| Goat Anti-Rabbit IgG H&L (HRP) | Antibody | Abcam | ab97051 | |
| Goat Anti-Mouse IgG H&L (HRP) | Antibody | Abcam | ab97023 | |

## Procedure

### Notes

- U-87 MG cells are maintained in DMEM supplemented with 10% FBS at 37°C/5% $CO_2$.
- All cells will be sent for mycoplasma testing and STR profiling.

1. Plate U-87 MG cells in 60 mm dishes.
2. 24 hr after plating, transfect cells with plasmids (maxiprepped in Protocol 1) using *Trans*IT-LT1 Transfection Reagent according to manufacturer's protocol.
   a. #Transfect 8 µg of DNA per construct using appropriate volume of transfection reagent.
      i. Empty vector.
      ii. IDH1$^{R132H}$ vector.
3. 48 hr after transfection, treat cells with vehicle or 5 mM Oct-α-KG for 6 hr.
   a. Vehicle is DMSO.
4. Wash cells once with cold PBS, then lyse cells in 0.5 ml of SDS loading buffer.
   a. #4 SDS-PAGE loading buffer: 50 mM Tris pH 6.8, 2% SDS, 10% glycerol, 1% B-ME, 12.5 mM EDTA, 0.02% bromophenol blue.
5. Heat lysates at 99°C for 10 min.
6. Run SDS-PAGE gel until ladder marker reaches the bottom of the gel.
7. #Equilibrate gel in transfer buffer for 15 min.
8. #Meanwhile, cut membrane and 4 pieces 3 MM filter paper to size of gel.
   a. Soak membrane in MeOH for a few seconds, then wash with $H_2O$.
   b. Soak membrane, 3 MM filter paper and pads in transfer buffer.
      i. Transfer buffer: 38 mM glycine, 47 mM Tris, 11 mM SDS, 20% MeOH.
9. #Assemble transfer cassette:
   a. red pole (+) < clear plate < pad < 2 × 3 MM filter paper < membrane < gel < 2 × 3 MM filter paper < pad < black pole (−).
10. #Add stirring bar and ice box to transfer box and fill box with transfer buffer until cassette is submerged.
    a. Run at 100 V for 1 hr.
11. #Wash membrane in wash buffer for 2 × 5 min.
    a. Wash buffer: 1× PBS with 0.05% Tween-20 and 0.1% sodium azide.
12. #Incubate membrane in blocking buffer for 30 min.
    a. Blocking buffer: 3% non-fat milk in PBS.
13. #Incubate membrane with primary antibody in blocking buffer for 2 hr at room temperature (RT) or overnight at 4°C (use manufacturer's suggested dilution in blocking buffer).
    a. H3.
    b. IDH1.
    c. H3K4me1.
    d. H3K4me3.
    e. H3K9me2.
    f. H3K27me2.
    g. H3K79me2.
14. #Wash 5 min 2× with wash buffer.
15. #Incubate membrane with secondary antibody for 90 min at RT (use manufacturer's suggested dilution in blocking buffer).
    a. HRP-conjugated Goat Anti-Mouse IgG H&L: 1:2000.
    b. HRP-conjugated Goat Anti-Rabbit IgG H&L: 1:2000.
16. #Wash 3 × 5 min in wash buffer.
17. # Detect HRP-conjugated secondary antibodies with chemiluminescent detection according to the manufacturer's protocol and image on the Typhoon scanner.
18. Strip the blot in between probes:
    a. Wash the membrane with 100 ml stripping buffer (100 mM betamercaptoethanol, 1% SDS 25 mM glycine pH 2.0) for 30 min with agitation.
    b. Wash the stripped membrane twice with Western blotting wash buffer, 600 ml each wash, for 10 min with agitation.
    c. Go to the blocking step of the western blot protocol.
    d. Check that stripping was successful by repeating the detection step (without re-probing). Record image of the stripped gel. This will confirm the first antibody-HRP conjugate is removed

and/or inactivated. If the stripping procedure is successful, wash the membrane with washing buffer and repeat the blocking-probing and detection steps for the second antibody.
i. Note: if stripping is unsuccessful, individual blots will be performed.
19. Quantify intensity of bands on western blots using ImageQuant 5.2. Normalize levels of methylated histones to total H3 protein level. Normalize IDH1$^{R132H}$ + vehicle and IDH1$^{R132H}$ + oct-α-KG treated samples to vector + vehicle samples for each normalized methylated histone.
20. Repeat independently 5 additional times.

## Deliverables

- Data to be collected:
  ○ Full scans of western blots for H3, IDH1, H3K4me1, H3K4me3, H3K9me2, H3K27me2, and H3K79me2 (as seen in Figure 3D) including ladder.
  ○ Raw values of intensity of western blot bands as measured by ImageQuant 5.2 software.
  ○ Quantification of methylated histone values normalized to total protein level.
  ○ Quantification of average values and standard deviations for each condition. Levels of methylated histone in vector control cells are set to 100% and levels of methylated histone for other conditions are relative to vector control.
  ○ Table of average ± standard deviation of methylated histone levels normalized to H3 for each condition and relative to vector control cells (as seen in Supplemental Figure 3J).

## Confirmatory analysis plan

- Statistical analysis of the replication data:
  ○ Note: At the time of analysis we will perform the Shapiro–Wilk test and generate a quantile–quantile plot to assess the normality of the data. We will also perform Levene's test to assess homoscedasticity. If the data appears skewed we will perform the appropriate transformation in order to proceed with the proposed statistical analysis. If this is not possible we will perform the equivalent non-parametric test.
  ○ One-way MANOVA of normalized H3K4me1, H3K4me3, H3K9me2, H3K27me2, and H3K79me2 levels from IDH1$^{R132H}$ + vehicle and IDH1$^{R132H}$ + oct-α-KG cells with the following Bonferroni corrected comparisons:
    ▪ H3K4me1 levels of IDH1$^{R132H}$ vs IDH1$^{R132H}$ + oct-α-KG.
    ▪ H3K4me3 levels of IDH1$^{R132H}$ vs IDH1$^{R132H}$ + oct-α-KG.
    ▪ H3K9me2 levels of IDH1$^{R132H}$ vs IDH1$^{R132H}$ + oct-α-KG.
    ▪ H3K27me2 levels of IDH1$^{R132H}$ vs IDH1$^{R132H}$ + oct-α-KG.
    ▪ H3K79me2 levels of IDH1$^{R132H}$ vs IDH1$^{R132H}$ + oct-α-KG.
  ○ Bonferroni corrected one-sample t-tests (outside the MANOVA framework) of normalized levels from IDH1$^{R132H}$ + vehicle of the following conditions compared to constant (vector + vehicle set to 100):
    ○ H3K4me1.
    ○ H3K4me3.
    ○ H3K9me2.
    ○ H3K27me2.
    ○ H3K79me2.
- Meta-analysis of original and replication attempt effect sizes:
  ○ Compute the effect sizes of each comparison, compare them against the reported effect size in the original paper and use a meta-analytic approach to combine the original and replication effects, which will be presented as a forest plot.

## Known differences from the original study

- While the manufacturer was specified for antibodies used, the exact catalog number was not. The RP:CB core team chose the most appropriate antibody from the manufacturer based on manufacturer's recommended applications and user reviews of the antibody.
- Aspects of the Western blot protocol are provided by the replicating lab; complete details of the original protocol were unavailable.

## Provisions for quality control
All data obtained from the experiment—raw data, data analysis, control data and quality control data—will be made publicly available, either in the published manuscript or as an open access dataset available on the Open Science Framework (https://osf.io/kvshc/).

- STR profiling confirming cell line authenticity.
- Mycoplasma testing confirming lack of contamination.
- Images of stripped gel membranes confirming stripping was successful.

## Protocol 4: Dot blot to measure of levels of 5hmC in genomic DNA

This protocol describes how to transfect HEK293 cells with vectors expressing the catalytic domain of TET2 (TET2-CD) and wild-type or mutant forms of IHD1 and IDH2 and then assess genomic DNA hydroxymethylation by dot blot, as seen in Figure 7B and Supplemental Figure 7C.

### Sampling

- This experiment will be conducted independently 4 times for a final power of 96%.
  ○ See Power calculations for details.
- Each experiment consists of 9 cohorts:
  ○ Cohort 1: Untransfected cells [additional control].
  ○ Cohort 2: Vector transfected cells.
  ○ Cohort 3: FLAG-TET2-CD transfected cells.
    ■ The catalytic domain of TET2.
  ○ Cohort 4: FLAG-TET2-CM transfected cells.
    ■ CM: mutant version of the TET2 catalytic domain.
  ○ Cohort 5: FLAG-TET2-CD + FLAG-IDH1 transfected cells.
  ○ Cohort 6: FLAG-TET2-CD + FLAG-IDH1$^{R132H}$ transfected cells.
  ○ Cohort 7: FLAG-TET2-CD + FLAG-IDH2 transfected cells.
  ○ Cohort 8: FLAG-TET2-CD + FLAG-IDH2$^{R140Q}$ transfected cells.
  ○ Cohort 9: FLAG-TET2-CD + FLAG-IDH2$^{R172K}$ transfected cells.
  ○ Each cohort will have gDNA spotted out at 5, 10, 25, 50, 100 and 250 ng and probed with anti-5hmC antibody.

### Materials and reagents

| Reagent | Type | Manufacturer | Catalog # | Comments |
|---------|------|-------------|-----------|----------|
| Mouse monoclonal IgG$_{2a}$ α Anti-5hmC | Antibody | Active Motif | 40000 | Original catalog number unspecified |
| Mouse monoclonal IgG1 α FLAG | Antibody | Sigma–Aldrich | F3165 | Original catalog number unspecified |
| Goat Anti-Mouse IgG H&L (HRP) | Antibody | Abcam | ab97023 | |
| HEK293 cells | Cells | ATCC | CRL-1573 | Original unspecified |
| Typhoon scanner | Equipment | Amersham/ GE Health Sciences | 9410 | |
| Hybond ECL nitrocellulose membranes; 20 cm × 20 cm | Reagent | GE Healthcare (Sigma–Aldrich) | GERPN2020D | Original not specified |
| DMEM; high glucose | Medium | Sigma–Aldrich | D5671 | Original unspecified |
| FBS | Reagent | Sigma–Aldrich | F2442 | Original unspecified |
| Vector alone | Plasmid | Provided by original authors | | |
| FLAG-TET2-CD | Plasmid | Provided by original authors | | |
| FLAG-TET2-CM | Plasmid | Provided by original authors | | |
| FLAG-IDH1 | Plasmid | Provided by original authors | | |
| FLAG-IDH1$^{R132H}$ | Plasmid | Provided by original authors | | |
| Flag-IDH2 | Plasmid | Provided by original authors | | |
| FLAG-IDH2$^{R140Q}$ | Plasmid | Provided by original authors | | |
| FLAG-IDH2$^{R172K}$ | Plasmid | Provided by original authors | | |
| TransIT-LT1 transfection reagent | Reagent | Mirus Bio | MIR 2300 | Replaces SunBio-EZ (SunBio) |
| Nonfat-dried milk bovine | Reagent | Sigma–Aldrich | M7409 | |

*Continued on next page*

*Continued*

| Reagent | Type | Manufacturer | Catalog # | Comments |
|---|---|---|---|---|
| ECL prime Western blotting system | Reagent | GE Healthcare (Sigma–Aldrich) | GERPN2232 | Original not specified |
| Image Quant 5.2 | Software | GE | Version 5.2 | |
| Protease inhibitor cocktail (mammalian) | Reagent | Sigma–Aldrich | P8340-1ML | Original not specified |
| TruPAGE TEA-Tricine SDS running buffer (20×) | Reagent | Sigma–Aldrich | PCG3001-500 ML | Original not specified |
| TruPAGE LDS sample buffer (4×) | Reagent | Sigma–Aldrich | PCG3009-10 ML | Original not specified |
| TruPAGE DTT sample reducer (10×) | Reagent | Sigma–Aldrich | PCG3005-1ML | Original not specified |
| TruPAGE transfer buffer (20×) | Reagent | Sigma–Aldrich | PCG3011-500 ML | Original not specified |
| PBS, without $MgCl_2$ and $CaCl_2$ | Reagent | Sigma–Aldrich | D8537 | Original not specified |
| Ponceau S solution; 0.1% (wt/vol) in 5% acetic acid | Reagent | Sigma–Aldrich | P7170 | Original not specified |
| Tris buffered saline (TBS); 10× solution | Reagent | Sigma–Aldrich | T5912 | Original not specified |
| Bradford reagent | Reagent | Sigma–Aldrich | B6916 | Original not specified |
| QIAamp DNA mini kit | Kit | Qiagen | 51304 | |

## Procedure

### Notes

- This protocol contains information from Ito and colleagues (*Ito et al., 2010*).
- HEK293 cells are maintained in DMEM supplemented with 10% FBS at 37°C/5% $CO_2$.
- All cells will be sent for mycoplasma testing and STR profiling.

1. Transform, grow up and maxiprep plasmids using an Endo-free Maxiprep kit following the manufacturer's instructions.
   a. Confirm plasmid identity by sequencing.
2. Plate $6 \times 10^5 - 1.2 \times 10^6$ HEK293 cells per 60 mm dish.
3. 24 hr after plating, transfect cells with indicated plasmids.
   a. #Transfect cells with 8 μg of DNA per construct using *Trans*IT-LT1 Transfection Reagent according to manufacturer's protocol#.
   i. Cohort 1: Untransfected cells.
   ii. Cohort 2: Vector only.
   iii. Cohort 3: FLAG-TET2-CD.
   iv. Cohort 4: FLAG-TET2-CM.
   v. Cohort 5: FLAG-TET2-CD + FLAG-IDH1.
   vi. Cohort 6: FLAG-TET2-CD + FLAG-IDH1$^{R132H}$.
   vii. Cohort 7: FLAG-TET2-CD + FLAG-IDH2.
   viii. Cohort 8: FLAG-TET2-CD + FLAG-IDH2$^{R140Q}$.
   ix. Cohort 9: FLAG-TET2-CD + FLAG-IDH2$^{R172K}$.
4. *For each cohort, transfect two parallel plates; harvest genomic DNA from one plate (proceed to Step 5) and protein from the second plate (proceed to Step 7).
5. 36–40 hr after transfection, isolate genomic DNA from cells on the first plate using the QIAamp kit according to the manufacturer's instructions.
   a. Determine DNA concentration and purity.
6. Dot blot to assess levels of 5hmC:
   a. Quantify gDNA concentration using a NanoDrop. #Spot genomic DNA onto nitrocellulose membrane using a pipet, then crosslink the DNA to the membrane by UV irradiation for 2 min.
   i. The following amounts of genomic DNA should be spotted: 250 ng, 100 ng, 50 ng, 25 ng, 10 ng, and 5 ng.
   b. Bake nitrocellulose membrane at 80°C for #1 hr.

 c. Block membrane with 5% skim milk in TBS with 0.1% Tween 20 (TBST) for 1 hr.

 d. Perform western blot on spotted nitrocellulose with the following antibody: anti-5hmC. Incubate membrane with primary antibody diluted 1:10,000 overnight at 4°C.

 e. Wash membrane three times with TBST.

 f. Incubate membrane with secondary antibody (HRP-conjugated anti-rabbit IgG) diluted 1:2000 for 1 hr at room temperature.

 g. Wash membrane three times with TBST, then treat with ECL and scan with a Typhoon scanner.

 h. Quantify dot-blot using Image-Quanta software.

7. Check expression of exogenous proteins by Western blot using the second plate.

 a. Wash cells once with cold PBS, then lyse cells in 0.5 ml of SDS loading buffer.

 i. #4× SDS-PAGE loading buffer: 50 mM Tris pH 6.8, 2% SDS, 10% glycerol, 1% B-ME, 12.5 mM EDTA, 0.02% bromophenol blue.

 b. Heat lysates at 99°C for 10 min.

 c. Run SDS-PAGE gel until ladder marker reaches the bottom of the gel.

 d. #Equilibrate gel in transfer buffer for 15 min.

 e. #Meanwhile, cut membrane and 4 pieces 3 MM filter paper to size of gel.

 i. Soak membrane in MeOH for a few seconds, then wash with $H_2O$.

 ii. Soak membrane, 3 mM filter paper and pads in transfer buffer.

 iii. Transfer buffer: 38 mM glycine, 47 mM Tris, 11 mM SDS, 20% MeOH

 f. #Assemble transfer cassette:

 i. red pole (+) < clear plate < pad < 2 × 3 MM filter paper < membrane < gel < 2 × 3 MM filter paper < pad < black pole (−).

 g. #Add stirring bar and ice box to transfer box and fill box with transfer buffer until cassette is submerged.

 i. Run at 100 V for 1 hr.

 h. #Wash membrane in wash buffer for 2 × 5 min.

 i. Wash buffer: 1× PBS with 0.05% Tween-20 and 0.1% sodium azide.

 i. #Incubate membrane in blocking buffer for 30 min.

 i. Blocking buffer: 3% non-fat milk in PBS.

 j. #Incubate membrane with primary antibody in blocking buffer for 2 hr at RT or O/N at 4°C (use manufacturer's suggested dilution in blocking buffer).

 i. α FLAG.

 k. #Wash 5 min 2× with wash buffer.

 l. #Incubate membrane with secondary antibody for 90 min at RT (use manufacturer's suggested dilution in blocking buffer).

 i. HRP-conjugated Goat Anti-Mouse IgG H&L: 1:2000.

 m. #Wash 3 × 5 min in wash buffer.

 n. # Detect HRP-conjugated secondary antibodies with chemiluminescent detection according to the manufacturer's protocol and image on the Typhoon scanner.

 o. Quantify intensity of dots on western blots using ImageQuant 5.2.

 i. Normalize values to FLAG-TET2-CD transfected cells.

8. Repeat independently three additional times.

## Deliverables

- Data to be collected:
  - Chromatograms and sequence files confirming plasmid identity.
  - DNA concentration and purity data.
  - Full scans of dot blots for anti-5hmC and western blots for anti-FLAG (as seen in Figure 7B).
  - Raw values of intensity of dot blot as measured by Image-Quanta software.
  - Quantification of 5hmc values relative to TET2-CD.
  - Quantification of average values and standard deviations for each condition for all experiments.
  - Bar graph and table of average values and standard deviations relative to TET2-CD samples (as seen in Figure 7B and Supplemental Figure 7C).

## Confirmatory analysis plan

- Statistical analysis of the replication data:

- ○ Note: At the time of analysis we will perform the Shapiro–Wilk test and generate a quantile–quantile plot to assess the normality of the data. We will also perform Levene's test to assess homoscedasticity. If the data appears skewed we will perform the appropriate transformation in order to proceed with the proposed statistical analysis. If this is not possible we will perform the equivalent non-parametric test.
  - ○ Comparison of the various genotypes for each of the DNA concentrations.
    - ■ Bonferonni corrected one-sample $t$-test of normalized 5hmC levels of the following cohorts compared to constant (TET2-CD set to 1):
      - TET-2CD + IDH1.
      - TET-2CD + IDH1$^{R132H}$.
      - TET-2CD + IDH2.
      - TET-2CD + IDH2$^{R140Q}$.
      - TET-2CD + IDH2$^{R172K}$.
- Meta-analysis of original and replication attempt effect sizes:
  - ○ Compute the effect sizes of each comparison, compare them against the reported effect size in the original paper and use a meta-analytic approach to combine the original and replication effects, which will be presented as a forest plot.

## Known differences from the original study

- Aspects of the Western blot protocol are provided by the replicating lab; complete details of the original protocol were unavailable.

## Provisions for quality control

All data obtained from the experiment—raw data, data analysis, control data and quality control data—will be made publicly available, either in the published manuscript or as an open access dataset available on the Open Science Framework (https://osf.io/kvshc/).

- Sequence data confirming plasmid identity.
- Western blots confirming exogenous protein expression.
- STR profiling confirming cell line authenticity.
- Mycoplasma testing confirming lack of contamination.

## Protocol 5: Radiolabeled 5mC-5hmC conversion assay

This protocol describes how to run the in vitro assay to examine the effect of 2-HG on the TET family of methyl hydroxylases, as seen in Figure 8A.

## Sampling

- This experiment will be performed independently a total of 6 times for a final power of ≥80%.
  - ○ The original data is qualitative, thus to determine an appropriate number of replicates to initially perform, sample sizes based on a range of potential variance was determined.
  - ○ See Power calculations for details.
- Each experiment consists of 8 cohorts:
  - ○ No recombinant protein.
  - ○ FLAG-TET2-CD + vehicle.
  - ○ FLAG-TET2-CD + 10 mM D-2-HG.
  - ○ FLAG-TET2-CD + 25 mM D-2-HG.
  - ○ FLAG-TET2-CD + 50 mM D-2-HG.
  - ○ FLAG-TET2-CD + 10 mM L-2-HG.
  - ○ FLAG-TET2-CD + 25 mM L-2-HG.
  - ○ FLAG-TET2-CD + 50 mM L-2-HG.
  - ○ Each cohort will detect:
    - ■ 5m-dCMP.
    - ■ 5hm-dCMP.

## Materials and reagents

| Reagent | Type | Manufacturer | Catalog # | Comments |
|---|---|---|---|---|
| D-2-HG | Reagent | Sigma–Aldrich | H8378 | |
| L-2-HG | Reagent | Sigma–Aldrich | 90790 | |
| Sf9 cells | Cells | ATCC | CRL-1711 | Original unspecified |
| Shrimp alkaline phosphatase | Reagent | New England Biolabs | MO371S | |
| T4 polynucleotide kinase | Reagent | Sigma–Aldrich | KEM0006 | |
| DNase I | Reagent | Sigma–Aldrich | AMPD1 | |
| Phosphodiesterase I | Reagent | Sigma–Aldrich | P3243 | |
| PEI-cellulose TLC plate | Material | Sigma–Aldrich | Z122882 | |
| FLAG-TET2-CD viral particles | Virus | Provided by the original authors | | |
| Anti-Flag M2 antibody agarose affinity gel | Reagent | Sigma–Aldrich | A2220 | |
| Flag peptide | Reagent | Sigma–Aldrich | F4799 | |
| α-KG | Reagent | Sigma–Aldrich | 75,892 | |
| GenElute PCR Clean-Up Kit | Kit | Sigma–Aldrich | NA1020-1KT | Replaces Qiagen cat no. 28304 |
| [γ-32]ATP | Reagent | Perkin Elmer | BLU502H/NEG502H | |
| MspI methyltransferase | Reagent | NEB | M0215L | |
| MspI restriction endonuclease | Reagent | NEB | R0106T | |
| DNA duplex oligonucleotide substrate | oligo | Integrated DNA Technologies | custom 5'-GTGTTCTTTCAGCTCCGGTCACGCTGACCAGC-3' as a duplex oligo, HPLC purified at 1 umole scale maybe higher depending on recovery | |
| M13-F primer | oligo | Integrated DNA Technologies | | CCAGTCACGACGTTGTAAAACG |
| M13-R primer | oligo | Integrated DNA Technologies | | CCAGTCACGACGTTGTAAAACG |
| JumpStart REDTaq DNA Polymerase | Reagent | Sigma | D8189-50UN | |
| dNTP mix 10 mM | Reagent | Sigma | D7295-.2 ML | |
| BlueView TAE buffer | Buffer | Sigma | T8935-1L | |
| Molecular biology grade water | Reagent | Sigma | W4502-1L | |

## Procedure

Note: This protocol contains information from Ito and colleagues (2010).

1. Generate recombinant FLAG-TET2-CD virus from supplied virus stock.
   a. #Infect a 5 ml culture with 0.1 ml of virus stock supplied.
      i. Grow in a stationary tissue culture flask at 27˚C.
   b. #After 5 days, collect the virus. Simultaneously, start a 50 ml suspension culture at 27˚C with 140 rpm shaking.
      i. Confirm viral insert identity by sequencing using M13F and R primers and REDTaq polymerase, followed by gel purification and sequencing of PCR product.
   c. #After culturing for 3 days, infected the suspension culture with 2.5 ml of virus stock.
   d. #4 days after infection collect virus. Simultaneously, start new 50 ml suspension cultures for protein expression.
      e. #After 3 days of culture, the suspension cultures are infected with 2.5 ml virus.
   e. #After 3 days of infection the cells expressing recombinant protein are collected by centrifugation and stored at −80˚C until the protein is to be purified.
      i. #More round of expression may be required depending on expression level.
   f. Purify baculovirus expressed recombinant FLAG-TET2-CD from insect Sf9 cells with anti-Flag M2 antibody agarose affinity gel and elute with buffer containing 10 mM Tris–HCl pH 8.0, 150 mM NaCl, 1 mM DTT, 15% glycerol and 0.2 µg/µl Flag peptide.
   g. Note; generate sufficient recombinant protein to use in a total of 6 replicates of this protocol.

2. #Prepare methylated oligonucleotide substrate.
   a. Treat unmethylated DNA duplex oligo with MspI methyltransferase for 2 hr at 37°C following manufacturer's instructions.
   b. Purify with a QiaQuick Nucleotide Removal kit following manufacturer's instructions.
3. Incubate 5 μg of the purified recombinant TET2-CD protein and various concentrations of vehicle only, D-2-HG, or L-2-HG with 0.5 μg methylated oligonucleotide substrate in vehicle (50 mM HEPES (pH 8), 75 μM Fe(NH$_4$)$_2$(SO$_4$)$_2$, 2 mM ascorbate) and 0.1 mM α-KG for 3 hr at 37°C.
   a. See cohorts for detailed concentrations to use.
   b. #If necessary, concentrate protein to ensure the final reaction volume is between 100–1000 μl.
   c. Purify oligonucleotide substrates using a GenElute PCR Clean-Up Kit following manufacture's instructions.
4. Digest oligonucleotides with 1 U/μg MspI restriction endonuclease at 37°C for #2 hr following manufacturer's instructions.
5. Treat digested DNA with 1U/μmol shrimp alkaline phosphatase at 37°C for #2 hr.
   a. #Heat inactivate at 65°C for 10 min.
6. Label DNA with [γ-32]ATP and polynucleotide kinase.
   a. #Add 1 μl of [γ-32]ATP at 3000 Ci/mmol, 5 mCi/ml and 1 μl polynucleotide kinase to the previous reaction.
   b. #Incubate for 1 hr at 37°C.
7. Ethanol precipitate labeled fragments.
   i. #Add 3M NaOAc to a final concentration of 0.3M.
   ii. #Add 2 vol 100% EtOH.
   iii. #Incubate mixture at on dry ice for 20 min.
   iv. #Centrifuge in a microfuge at 4°C at maximum speed for 10 min.
   v. #Remove supernatant and air dry pellet.
   vi. #Resuspend.
8. Digest labeled fragments with 10 μg DNAse I and 10 μg phosphodiesterase I in the presence of 15 mM MgCl$_2$ and 2 mM CaCl$_2$ at 37 °C for #2 hr.
9. Spot 1 μl of digestion product from step 8 onto a PEI-cellulose TLC plate and separate in an isobutyric acid/water/ammonium hydroxide (66:20:2) buffer.
10. Dry the TLC plate and then expose to film.
11. Quantify intensity of 5hmC bands.
    a. Normalize values to FLAG-TET2-CD + vehicle.
12. Repeat independently five additional times starting at Step 2.

## Deliverables

- Data to be collected:
  ○ Sequencing data confirming viral insert identity.
  ○ Data about viral titer and amount of and quality of protein generated.
  ○ Scans of films exposed to TLC plate (as in Figure 8A, left).
  ○ Raw values of intensity of 5hm-dCMP (5hmC) spots.
  ○ Quantification of 5hmC intensity relative to FLAG-TET2-CD (recombinant protein) + vehicle sample.
  ○ Quantification of average values and standard deviations for each condition for triplicate experiments.
  ○ Bar graph of relative 5hmC intensity for each sample with standard deviations (as in Figure 8A, right).

## Confirmatory analysis plan

- Statistical Analysis of the Replication Data:
  ○ Note: At the time of analysis we will perform the Shapiro–Wilk test and generate a quantile–quantile plot to assess the normality of the data. We will also perform Levene's test to assess homoscedasticity. If the data appears skewed we will perform the appropriate transformation in order to proceed with the proposed statistical analysis. If this is not possible we will perform the equivalent non-parametric test.
  ○ Two-way ANOVA of normalized 5hmC levels of TET2-CD protein treated with D-2-HG or L-2-HG with the following Bonferroni corrected comparisons:
    - 10 mM D-2-HG vs 10 mM L-2-HG.
    - 50 mM D-2-HG vs 50 mM L-2-HG.

- 10 mM D-2-HG vs 50 mM D-2-HG.
  ○ Bonferroni corrected one-sample *t*-tests (outside the ANOVA framework) of normalized 5hmC levels of TET2-CD protein treated with the following concentrations of D-2-HG compared to constant (TET2-CD + vehicle set to 1):
    - 10 mM D-2-HG.
    - 50 mM D-2-HG.
- Meta-analysis of original and replication attempt effect sizes:
  ○ The replication data (mean and 95% confidence interval) will be plotted with the original reported data value plotted as a single point on the same plot for comparison.

## Known differences from the original study

- The lab provided the protocol for expansion of the viral aliquot shared by the original authors for generation of the recombinant FLAG-TET2 protein.

## Provisions for quality control

All data obtained from the experiment—raw data, data analysis, control data and quality control data—will be made publicly available, either in the published manuscript or as an open access dataset available on the Open Science Framework (https://osf.io/kvshc/).

- Sequence data confirming viral insert identity.
- Data about viral titer and amount of and quality of protein generated.

### Power calculations

Power calculations are performed to calculate the number of samples required to achieve at least 80% power and the indicated alpha error. For a detailed breakdown of all power calculations, please see spreadsheet at https://osf.io/gnsti/wiki/home/.

### Protocol 1

### Summary of original data

- Note: Data estimated from published figures.

| Supp. Figure 3I: Levels of α-KG with WT or mutant IDH1 | Mean | SD | N |
|---|---|---|---|
| Vector-transfected U-87 MG cells | 84 | 0* | 2 |
| WT IDH-transfected U-87 MG cells | 120 | 7.8 | 2 |
| IDH$^{R132H}$-transfected U-87 MG cells | 41 | 14 | 2 |

| Supp. Figure 3I: Levels of 2-HG with WT or mutant IDH1. | Mean | SD | N |
|---|---|---|---|
| Vector-transfected U-87 MG cells | 90 | 0* | 2 |
| WT IDH-transfected U-87 MG cells | 140 | 0* | 2 |
| Mutant IDH-transfected U-87 MG cells | 1730 | 14 | 2 |

*Because the original data reported null variances, the calculations below used the average of the non-null variances, 11.9, in place of a SD of 0.

### Test family

- Due to a lack of raw original data, we are unable to perform power calculations using a MANOVA. We are determining sample size calculations using a two-way ANOVA.
- Two-way ANOVA followed by Bonferroni corrected comparisons.

## Power calculations

- Calculations were performed with R software, version 3.1.2 (*R Core Team, 2014*) and G*Power software, version 3.1.7 (*Faul et al., 2007*).

**ANOVA calculations; α = 0.05**

| F(1,6) metabolite | Partial η2 | Effect size *f* | Power | Total sample size |
|---|---|---|---|---|
| 6702.3 | 0.999106 | 33.43005 | 99.9% | 7* |

*With 5 samples per group (30 samples total), power achieved is 99.9%.

**Corrected *t*-test sample size calculations; α = 0.0083333**

| | Group 1 | Group 2 | Effect size *d* | Power | Sample size per group |
|---|---|---|---|---|---|
| α-KG | Vector | IDH1-WT | 3.57815 | 80.1%* | 4* |
| | Vector | IDH1-R132H | 3.30960 | 92.0% | 5 |
| | IDH1-WT | IDH1-R132H | 6.97125 | 97.4%† | 3† |
| 2HG | Vector | IDH1-WT | 4.20168 | 93.1%‡ | 4‡ |
| | Vector | IDH1-R132H | 126.22669 | 99.9%§ | 2§ |
| | IDH1-WT | IDH1-R132H | 122.37832 | 99.9%# | 2# |

*With a sample size of 5 per group, the achieved power is 95.7%.
†With a sample size of 5 per group, the achieved power is 99.9%.
‡With a sample size of 5 per group, the achieved power is 99.2%.
§With a sample size of 5 per group, the achieved power is 99.9%.
#With a sample size of 5 per group, the achieved power is 99.9%.

## Sensitivity calculations

- Comparing 2-HG levels from Vector to IDH1 WT:
  - Based on a sample size of 4 per group, we will be able to see an effect size of 3.3710662 with α = 0.01 and a power of 80%.

## Protocol 2

## Summary of original data

- Note: Data estimated from published figures.

| **Supp. Fig. 3F: Quantification of Figure 3A Western Blots** | | **Mean** | **SD** | **N** |
|---|---|---|---|---|
| Untreated cells | H3K9me2/H3 ratio | 1 | 0 | 3 |
| | H3K79me2/H3 ratio | 1 | 0 | 3 |
| 10 mM oct-2-HG treated cells | H3K9me2/H3 ratio | 3.8 | 0.5 | 3 |
| | H3K79me2/H3 ratio | 8.5 | 1.5 | 3 |
| 20 mM oct-2-HG treated cells | H3K9me2/H3 ratio | 5.5 | 0.3 | 3 |
| | H3K79me2/H3 ratio | 17.2 | 2.4 | 3 |
| 20 mM oct-2-HG + 5 mM oct-α-KG treated cells | H3K9me2/H3 ratio | 0.6 | 0.3 | 3 |
| | H3K79me2/H3 ratio | 0.9 | 0.3 | 3 |

## Test family

- Due to a lack of raw original data, we are unable to perform power calculations using a MANOVA. We are determining sample size calculations using a two-way ANOVA.
- Two-way ANOVA followed by Bonferroni corrected comparisons.

## Power calculations

- Calculations were performed with R software, version 3.1.2 (*R Core Team, 2014*) and G*Power software, version 3.1.7 (*Faul et al., 2007*).

**ANOVA calculations; $\alpha = 0.05$**

| F(1,16) histone | Partial $\eta2$ | Effect size $f$ | A priori power | Total sample size |
|---|---|---|---|---|
| 235.0200 | 0.936260 | 3.83259 | 99.9%* | 10* |

*With 3 samples per group (12 total), achieved power is 99.9%.

**Corrected $t$-tests sample size calculations; $\alpha = 0.0083$**

| | Group 1 | Group 2 | Effect size $d$ | Power | Sample size per group |
|---|---|---|---|---|---|
| H3K9me2 | Vehicle treated cells | 10 mM Oct-2-HG treated cells | 11.05934 | 99.9% | 3 |
| H3K79me2 | | | 8.92288 | 99.9% | 3 |
| H3K9me2 | Vehicle treated cells | 20 mM Oct-2-HG treated cells | 23.55408 | 99.9% | 3 |
| H3K79me2 | | | 14.24069 | 99.9% | 3 |
| H3K9me2 | 20 mM Oct-2-HG treated cells | 20 mM Oct-2-HG + 5 mM oct-$\alpha$-KG treated cells | 28.82353 | 99.9% | 3 |
| H3K79me2 | | | 16.46129 | 99.9% | 3 |

## Test family

- This is an additional analysis to allow a direct comparison with the original study.
- Bonferroni corrected one-sample $t$-tests compared to 1 (vehicle treated cells).

## Power calculations

- Calculations were performed with G*Power software, version 3.1.7 (*Faul et al., 2007*).

**Bonferroni corrected t-tests; $\alpha = 0.0083$**

| | Group | Constant | Effect size d | A Priori power | Sample size per group |
|---|---|---|---|---|---|
| H3K9me2 | 10 mM Oct-2-HG treated cells | 1 | 9.65517 | 90.3% | 3 |
| H3K79me2 | | | 8.62069 | 84.4% | 3 |
| H3K9me2 | 20 mM Oct-2-HG treated cells | 1 | 26.47059 | 99.9% | 3 |
| H3K79me2 | | | 11.65468 | o 96.6% | 3 |

## Protocol 3

## Summary of original data

- Note: Data estimated from published figure.

| Supp. Figure 3J: quantification of Western blot band intensities from Figure 3D normalized to vector control | Mean | SD | N |
|---|---|---|---|
| With vector + vehicle | | | |
| H3K4me1/H3 ratio | 100 | Unspecified | 3 |
| H3K4me3/H3 ratio | 100 | unspecified | 3 |
| H3K9me3/H3 ratio | 100 | unspecified | 3 |
| H3K27me2/H3 ratio | 100 | unspecified | 3 |
| H3K79me2/H3 ratio | 100 | unspecified | 3 |
| With IDH1$^{R132H}$ + vehicle | | | |
| H3K4me1/H3 ratio | 209 | 36 | 3 |
| H3K4me3/H3 ratio | 466 | 64 | 3 |
| H3K9me3/H3 ratio | 283 | 56 | 3 |
| H3K27me2/H3 ratio | 232 | 24 | 3 |
| H3K79me2/H3 ratio | 267 | 47 | 3 |
| With IDH1$^{R132H}$ and oct-α-KG | | | |
| H3K4me1/H3 ratio | 105 | 16 | 3 |
| H3K4me3/H3 ratio | 274 | 25 | 3 |
| H3K9me3/H3 ratio | 126 | 21 | 3 |
| H3K27me2/H3 ratio | 99 | 9 | 3 |
| H3K79me2/H3 ratio | 130 | 20 | 3 |

## Test family

- Due to a lack of raw original data, we are unable to perform power calculations using a MANOVA. We are determining sample size calculations using a two-way ANOVA.
- Two-way ANOVA followed by Bonferroni corrected comparisons.

## Power calculations

- Calculations were performed with R software, version 3.1.2 (*R Core Team, 2014*) and G*Power software, version 3.1.7 (*Faul et al., 2007*).

**ANOVA calculations; α = 0.05**

| F(1,20) cell treatments | Partial η2 | effect size $f$ | Power | Total Sample size |
|---|---|---|---|---|
| 119.5629 | 0.85670 | 2.44502 | 97.1%* | 12* |

*With 6 samples per group (for a total of 60 samples), the power achieved is 99.9%.

**Corrected *t*-test sample size calculations; α = 0.005**

| Group 1 | Group 2 | Histone | Effect size *d* | Power | Sample size per group |
|---|---|---|---|---|---|
| IDH1[R132H] + vehicle | IDH1[R132H] + oct-α-KG | H3K4me1/H3 ratio | 3.73338 | 94.2%* | 5* |
| | | H3K4me3/H3 ratio | 3.95184 | 82.1%† | 4† |
| | | H3K9me3/H3 ratio | 3.71240 | 93.9%‡ | 5‡ |
| | | H3K27me2/H3 ratio | 7.33811 | 95.0%§ | 3§ |
| | | H3K79me2/H3 ratio | 3.79314 | 94.9%# | 5# |

*With a sample size of 6 per group, the achieved power is 98.9%.
†With a sample size of 6 per group, the achieved power is 99.5%.
‡With a sample size of 6 per group, the achieved power is 98.8%.
§With a sample size of 6 per group, the achieved power is 99.9%.
#With a sample size of 6 per group, the achieved power is 99.1%.

## Test family

- Outside the ANOVA framework
- Bonferroni corrected one-sample *t*-tests compared to 1 (vector + vehicle).

## Power calculations

- Calculations were performed with G*Power software, version 3.1.7 (*Faul et al., 2007*).

**Corrected *t*-test sample size calculations; α = 0.005**

| Group 1 | Constant | Histone | Effect size *d* | Power | Sample size per group |
|---|---|---|---|---|---|
| IDH1[R132H] + vehicle | 100 | H3K4me1/H3 ratio | 3.02778 | 94.2% | 6 |
| | | H3K4me3/H3 ratio | 5.71875 | 92.2%* | 4* |
| | | H3K9me3/H3 ratio | 3.26786 | 82.9%† | 5† |
| | | H3K27me2/H3 ratio | 5.50000 | 90.2%‡ | 4‡ |
| | | H3K79me2/H3 ratio | 3.55319 | 88.8%§ | 5§ |

*With a sample size of 6 per group, the achieved power is 99.9%.
†With a sample size of 6 per group, the achieved power is 96.9%.
‡With a sample size of 6 per group, the achieved power is 99.9%.
§With a sample size of 6 per group, the achieved power is 98.7%.

## Protocol 4

## Summary of original data

○ Note: Values estimated from published figure.

| Figure 7B: Relative 5hmC intensity | Mean | SD | N |
|---|---|---|---|
| **50 ng Genomic DNA** | | | |
| Vector | 0 | 0.01 | 3 |
| TET2-CD | 1 | 0 | 3 |
| TET2-CM | 0 | 0.01 | 3 |
| TET2-CD + IDH1 | 2.5 | 0.3 | 3 |

*Continued on next page*

*Continued*

**Figure 7B: Relative 5hmC**

| intensity | Mean | SD | N |
|---|---|---|---|
| **50 ng Genomic DNA** | | | |
| TET2-CD + IDH1$^{R132H}$ | 0.29 | 0.1 | 3 |
| TET2-CD + IDH2 | 2.6 | 0.11 | 3 |
| TET2-CD + IDH2$^{R40Q}$ | 0.31 | 0.07 | 3 |
| TET2-CD + IDH2$^{R172K}$ | 0.31 | 0.09 | 3 |

## Test family

- Bonferroni corrected one-sample *t*-tests compared to 1 (TET2-CD).

## Power calculations

- Power calculations were performed using G*Power software, version 3.1.7 (*Faul et al., 2007*).

**Corrected *t*-test sample size calculations; α = 0.01**

| Group 1: TET2 + | Constant | Effect size *d* | Power | Sample size per group |
|---|---|---|---|---|
| IDH1 | 1 | 5.00000 | 95.9% | 4 |
| IDH1$^{R132H}$ | 1 | 7.10000 | 99.9% | 4 |
| IDH2 | 1 | 14.54545 | 99.8%* | 3* |
| IDH2$^{R140Q}$ | 1 | 9.85714 | 94.6%† | 3† |
| IDH2$^{R172K}$ | 1 | 7.66667 | 82.9%‡ | 3‡ |

*With a sample size of 4 per group, the achieved power is 99.9%.
†With a sample size of 4 per group, the achieved power is 99.9%.
‡With a sample size of 4 per group, the achieved power is 99.9%.

# Protocol 5

## Summary of original data

- Note: Data estimated from published figures.

| Figure 8A: TLC blot intensities | Mean |
|---|---|
| TET2 + vehicle | 1 |
| TET2 + 10 mM D-2-HG | 0.67 |
| TET2 + 25 mM D-2-HG | 0.45 |
| TET2 + 50 mM D-2-HG | 0.17 |
| TET2 + 10 mM L-2-HG | 0.05 |
| TET2 + 25 mM L-2-HG | 0.03 |
| TET2 + 50 mM L-2-HG | 0.03 |

## Test family

- One way ANOVA followed by Bonferroni corrected comparisons.
- Outside the ANOVA framework
  - Bonferroni corrected one-sample *t*-tests compared to 1 (TET2 + vehicle).

## Power calculations

- Because the original data presented does not have variance (s.e.m. or s.d.), we have performed power calculations using several different levels of calculated variance and an assumed number of replicates to determine a suitable number of replications to perform.
- Calculations were performed with R software, version 3.1.2 (*R Core Team, 2014*) and G*Power software, version 3.1.7 (*Faul et al., 2007*).

**Calculated variances and assumed N**

| Figure 8A: dot blot intensities | Mean | N | 2% | 15% | 28% | 40% |
|---|---|---|---|---|---|---|
| TET2 + vehicle | 1 | 3 | n/a* | n/a* | n/a* | n/a* |
| TET2 + 10 mM D-2-HG | 0.67 | 3 | 0.0134 | 0.1005 | 0.1876 | 0.268 |
| TET2 + 25 mM D-2-HG | 0.45 | 3 | 0.009 | 0.0675 | 0.126 | 0.18 |
| TET2 + 50 mM D-2-HG | 0.17 | 3 | 0.0034 | 0.0255 | 0.0476 | 0.068 |
| TET2 + 10 mM L-2-HG | 0.05 | 3 | 0.001 | 0.0075 | 0.014 | 0.02 |
| TET2 + 25 mM L-2-HG | 0.03 | 3 | 0.0006 | 0.0045 | 0.0084 | 0.012 |
| TET2 + 50 mM L-2-HG | 0.03 | 3 | 0.0006 | 0.0045 | 0.0084 | 0.012 |

*Because each replicate will be normalized to TET2 + vehicle this will not have a variance associated with it. And thus the TET2 + vehicle is also not include in the ANOVA calculation.

## 2% variance

**ANOVA calculations; $\alpha = 0.05$**

| F(2,12) interaction | Partial $\eta 2$ | Effect size $f$ | Power | Total sample size |
|---|---|---|---|---|
| 1910.6 | 0.99687 | 17.8434 | 98.2%* | 9* |

*With 12 total samples, the power achieved is 99.9%.

**Corrected *t*-test sample size calculations; $\alpha = 0.01$**

| Group 1 | Group 2 | Effect size $d$ | Power | Sample size per group |
|---|---|---|---|---|
| 10 mM D-2-HG | 10 mM L-2-HG | 65.25231 | 99.9% | 2 |
| 50 mM D-2-HG | 50 mM L-2-HG | 57.34623 | 99.9% | 2 |
| 10 mM D-2-HG | 50 mM D-2-HG | 51.14839 | 99.9% | 2 |

**Corrected *t*-test sample size calculations; $\alpha = 0.01$**

| Group 1: | Constant | Effect size $d$ | Power | Sample size per group |
|---|---|---|---|---|
| 10 mM D-2-HG | 1 | 24.62687 | 97.7% | 2 |
| 50 mM D-2-HG | 1 | 244.11765 | 99.9% | 2 |

## 15% variance

**ANOVA calculations; $\alpha = 0.05$**

| F(2,12) interaction | Partial η2 | Effect size f | Power | Total sample size |
|---|---|---|---|---|
| 2.37930 | 0.84987 | 2.37930 | 93.8% | 12* |

*With 12 total samples, the power achieved is 99.9%.

**Corrected *t*-test sample size calculations; $\alpha = 0.01$**

| Group 1 | Group 2 | Effect size d | Power | Sample size per group |
|---|---|---|---|---|
| 10 mM D-2-HG | 10 mM L-2-HG | 8.70031 | 99.9% | 3 |
| 50 mM D-2-HG | 50 mM L-2-HG | 7.64616 | 99.4% | 3 |
| 10 mM D-2-HG | 50 mM D-2-HG | 6.81978 | 97.9% | 3 |

**Corrected *t*-test sample size calculations; $\alpha = 0.01$**

| Group 1: | Constant | Effect size d | Power | Sample size per group |
|---|---|---|---|---|
| 10 mM D-2-HG | 1 | 3.28358 | 87.2% | 6 |
| 50 mM D-2-HG | 1 | 32.54902 | 99.5% | 3 |

## 28% variance

**ANOVA calculations; $\alpha = 0.05$**

| F(2,12) interaction | Partial η2 | Effect size f | Power | Total sample size |
|---|---|---|---|---|
| 9.7548 | 0.61916 | 1.27507 | 86.1% | 12 |

**Corrected t-test sample size calculations; $\alpha = 0.01$**

| Group 1 | Group 2 | Effect size d | Power | Sample size per group |
|---|---|---|---|---|
| 10 mM D-2-HG | 10 mM L-2-HG | 4.66088 | 97.9% | 4 |
| 50 mM D-2-HG | 50 mM L-2-HG | 4.09616 | 93.6% | 4 |
| 10 mM D-2-HG | 50 mM D-2-HG | 3.65346 | 86.6% | 4 |

**Corrected *t*-test sample size calculations; $\alpha = 0.01$**

| Group 1: | Constant | Effect size d | Power | Sample size per group |
|---|---|---|---|---|
| 10 mM D-2-HG | 1 | 1.75906 | 80.3% | 14 |
| 50 mM D-2-HG | 1 | 17.43697 | 94.0% | 4 |

## 40% variance

**ANOVA calculations; $\alpha = 0.05$**

| F(2,12) interaction | Partial η2 | Effect size f | Power | Total sample size |
|---|---|---|---|---|
| 4.7765 | 0.44323 | 0.892237 | 82.2% | 17* |

*With 18 total samples, the power achieved is 85.3%.

**Corrected *t*-test sample size calculations; α = 0.01**

| Group 1 | Group 2 | Effect size *d* | Power | Sample size per group |
|---|---|---|---|---|
| 10 mM D-2-HG | 10 mM L-2-HG | 3.26262 | 92.8% | 5 |
| 50 mM D-2-HG | 50 mM L-2-HG | 2.86731 | 83.9% | 5 |
| 10 mM D-2-HG | 50 mM D-2-HG | 2.55742 | 86.3% | 6 |

**Corrected *t*-test sample size calculations; α = 0.01**

| Group 1: | Constant | Effect size *d* | Power | Sample size per group |
|---|---|---|---|---|
| 10 mM D-2-HG | 1 | 1.23134 | 80.8% | 27 |
| 50 mM D-2-HG | 1 | 12.20588 | 84.6% | 5 |

In order to produce quantitative replication data, we will run the experiment six times. Each time we will quantify band intensity. We will determine the standard deviation of band intensity across the biological replicates and combine this with the reported value from the original study to simulate the original effect size. We will use this simulated effect size to determine the number of replicates necessary to reach a power of at least 80%. We will then perform additional replicates, if required, to ensure that the experiment has more than 80% power to detect the original effect.

# Acknowledgements

The Reproducibility Project: Cancer Biology core team would like to thank the original authors, in particular Dr. Yue Xiong, for generously sharing reagents to ensure the fidelity and quality of this replication attempt. We would also like to thanks the following companies for generously donating reagents to the Reproducibility Project: Cancer Biology; American Tissue Culture Collection (ATCC), Applied Biological Materials, BioLegend, Charles River Laboratories, Corning Incorporated, DDC Medical, EMD Millipore, Harlan Laboratories, LI-COR Biosciences, Mirus Bio, Novus Biologicals, Sigma–Aldrich, and System Biosciences (SBI).

# Additional information

### Group author details

**Reproducibility Project: Cancer Biology**

Elizabeth Iorns: Science Exchange, Palo Alto, California; William Gunn: Mendeley, London, United Kingdom; Fraser Tan: Science Exchange, Palo Alto, California; Joelle Lomax: Science Exchange, Palo Alto, California; Timothy Errington: Center for Open Science, Charlottesville, Virginia

### Competing interests

RP:CB: We disclose that EI, FT, and JL are employed by and hold shares in Science Exchange Inc. The experiments presented in this manuscript will be conducted by BE at the Proteomics and Mass Spectrometry Facility, which is a Science Exchange lab. The other authors declare that no competing interests exist.

### Funding

| Funder | Author |
|---|---|
| Laura and John Arnold Foundation | Reproducibility Project: Cancer Biology |

The Reproducibility Project: Cancer Biology is funded by the Laura and John Arnold Foundation, provided to the Center for Open Science in collaboration with Science Exchange. The funder had no role in study design or the decision to submit the work for publication.

### Author contributions

BE, EG, Drafting or revising the article; RP:CB, Conception and design, Drafting or revising the article

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
