## [Decision Letter]

Thank you for submitting your work entitled “Registered report: Oncometabolite 2-Hydroxyglutarate Is a Competitive Inhibitor of α-Ketoglutarate-Dependent Dioxygenases” for peer review at *eLife*. Your submission has been favorably evaluated by Michael Marletta (Senior editor), Irwin Davidson (Reviewing editor), and four reviewers.

The reviewers have discussed the reviews with one another and the Reviewing editor has drafted this decision to help you prepare a revised submission. The reviewers of this paper have raised several issues and we would ask you to specifically take into account the comments concerning the statistical analyses.

Summary:

The article outlines the detailed protocol to reproduce a report published in *Cancer* Cell linking mutations in IDH1/2 to cellular levels of Ketoglutarate and changes in histone and DNA methylation. This initial work had a major impact on linking metabolites to chromatin, but also raised a number of questions that justify a rigorous replication.

Overall, the proposed study covers the major aspects with the required detail and rigour.

Specific points to address are:

1) The referees suggest that the authors consider using mass spectrometry to measure 5hmC in addition to immune dot-blot. Mass spectrometry is a more quantitative measure and while it would go beyond replicating the published findings it might give a clearer answer.

2) In protocol 1 a 2-way ANOVA is proposed, however as 2 quantitative variables are measured and there is only one qualitative factor (with three possible values) influencing these measures, an MANOVA would be more suited.

3) There may be confusion between groups and variables in setting the degrees of freedom for ANOVA analyses. In protocol 1, how was (2, 6) obtained? The same question applies to protocols 3, 4 and 5.

4) In addition to *t*-tests for the comparison of means where both variances are equal, F-tests should be added when variances are significantly different.

5) Referees raised concerns about null variances that appear in the power calculation tables. Although these values are not always available, variance values can change the conclusion of the tests. When variances are not available, preliminary experiments in order to estimate them are proposed. More generally, variance values used in this paper are estimated from published figures using a low number of replicates, so they are not robust. A way to increase robustness would be to increase measured values by a pre-determined factor and then relax the expected power if too many replicates are required.

Also, non-rounded computed sample sizes are requested to have an idea of how close we are to the theoretical value after rounding.

6) For protocol 2, in subsection “Confirmatory analysis plan”, a MANOVA is proposed, whereas a one-way ANOVA is suggested for the same protocol in subsection “Test family”. Please correct this inconsistency.

7) Protocol 5: a mean across several groups is compared with the mean of a single group. This should not be done with a simple *t*-test to take into account the fact that the number of measures is different in the two groups being compared.

---

## [Author Response]

*1) The referees suggest that the authors consider using mass spectrometry to measure 5hmC in addition to immune dot-blot. Mass spectrometry is a more quantitative measure and while it would go beyond replicating the published findings it might give a clearer answer*.

We agree mass spectrometry analysis would be a more quantitative approach to measure 5hmC levels, however feel it is beyond the scope of this project, which is to perform a direct replication of the original experiment(s). Aspects of an experiment not included in the original study are occasionally added to ensure the quality of the research, but by no means is a requirement of this project; rather, it is an extension of the original work. We know that the exclusion of certain experiments limits the scope of what can be analyzed by the project, but we are attempting to identify a balance of breadth of sampling for general inference with sensible investment of resources on replication projects to determine to what extent the included experiments are reproducible.

2) In protocol 1 a 2-way ANOVA is proposed, however as 2 quantitative variables are measured and there is only one qualitative factor (with three possible values) influencing these measures, an MANOVA would be more suited.

We agree and have included this in the confirmatory analysis section. However, as we do not have the raw data needed to perform a power calculation for the MANOVA test, we performed it with a 2-way ANOVA to estimate the needed sample size and adjusted the alpha error for the planned contrasts that will be performed to ensure the sample size is sufficient.

*3) There may be confusion between groups and variables in setting the degrees of freedom for ANOVA analyses. In protocol 1, how was (2, 6) obtained? The same question applies to protocols 3, 4 and 5*.

Thank you for catching these inconsistencies, we have checked and adjusted each protocol in regards to degrees of freedom. Some of the analysis sections have changed to address other questions below, but a link to all scripts has been provided below and in the revised manuscript.

*4) In addition to* t*-tests for the comparison of means where both variances are equal, F-tests should be added when variances are significantly different*.

We have added a note in the analysis section that at the time of analysis, we will assess the normality and homoscedasticity of the data. If necessary, we will perform the appropriate transformation in order to proceed with the proposed statistical analysis. We will note any changes or transformations made. We have updated the manuscript to address this point.

*5) Referees raised concerns about null variances that appear in the power calculation tables. Although these values are not always available, variance values* can *change the conclusion of the tests. When variances are not available, preliminary experiments in order to estimate them are proposed. More generally, variance values used in this paper are estimated from published figures using a low number of replicates, so they are not robust. A way to increase robustness would be to increase measured values by a pre-determined factor and then relax the expected power if too many replicates are required*.

*Also, non-rounded computed sample sizes are requested to have an idea of how close we are to the theoretical value after rounding*.

We agree about the concern about null variances and have reanalyzed the proposed analysis plans and power calculations to reflect this. In many cases the reason for the null variance was due to normalization to a common factor within each replicate – thus making the variance zero on purpose. In some cases we will need to repeat this as well (protocol 3, 4, and 5). In other cases (protocol 1 and 2) we used the other variances that were reported as an estimate for the null variances. And where possible (protocol 2) we included additional analysis to allow a direct comparison to the original analysis.

Full details of all power calculations are available through the study’s page on the Open Science Framework (https://osf.io/gnsti/?view_only=f3a48d5a355f429fa2264ee7c17e9705). Unfortunately, the program we use to calculate sample sizes, G*Power, only returns whole integers for recommended sample sizes.

*6) For protocol 2, in subsection “Confirmatory analysis plan”, a MANOVA is proposed, whereas a one-way ANOVA is suggested for the same protocol in subsection “Test family”. Please correct this inconsistency*.

The Confirmatory analysis plan is in reference to the proposed statistical analyses of the replication data. However, as we do not have the raw data needed to perform a power calculation for the MANOVA test, we performed it with a 2-way ANOVA to estimate the needed sample size and adjusted the alpha error for the planned contrasts that will be performed to ensure the sample size is sufficient.

*7) Protocol 5: a mean across several groups is compared with the mean of a single group. This should not be done with a simple* t*-test to take into account the fact that the number of measures is different in the two groups being compared*.

We had originally intended to perform a weighted planned contrast by comparing several groups to a single group – and agree an F test is properly suited. However, in the revised manuscript we are including multiple independent *t*-tests and one-sample *t*-tests (due to the necessary normalization described in point 5 above) to more thoroughly analyze the data as originally reported and interpreted.